Gluconobacter dominates the gut microbiome of the Asian palm civet Paradoxurus hermaphroditus that produces kopi luwak

Watanabe Hikaru 1
Ng Chong Han 2
http://orcid.org/0000-0001-5344-0585 Limviphuvadh Vachiranee 3
http://orcid.org/0000-0002-1816-4260 Suzuki Shinya 1
Yamada Takuji 1 takuji@bio.titech.ac.jp
1 School of Life Science and Technology, Tokyo Institute of Technology , Meguro, Tokyo , Japan
2 Faculty of Information Science & Technology, Multimedia University , Jalan Ayer Keroh Lama, Melaka , Malaysia
3 Bioinformatics Institute (BII), Agency for Science, Technology and Research (A*STAR) , Singapore , Singapore
Gillespie Joseph
Electronic publication date: 2020 Jul 30
Publication date: 2020
Volume: 8
Electronic Location ID: e9579
Received 2020 Mar 11; Accepted 2020 Jun 30
Copyright: © 2020 Watanabe et al.
Copyright year: 2020
Copyright holder: Watanabe et al.
License: This is an open access article distributed under the terms of the Creative Commons Attribution License, which permits unrestricted use, distribution, reproduction and adaptation in any medium and for any purpose provided that it is properly attributed. For attribution, the original author(s), title, publication source (PeerJ) and either DOI or URL of the article must be cited.
License URL: https://creativecommons.org/licenses/by/4.0/

Keywords: Microbiome, Fermentation, Kopi luwak

Funding: Education Academy of Computational Life Sciences, Tokyo Institute of Technology and Fundamental Research Grant Scheme MMUE.140076 Malaysia Ministry of Higher Education (MOHE) This work was supported by the Education Academy of Computational Life Sciences, Tokyo Institute of Technology and Fundamental Research Grant Scheme (grant number: MMUE.140076, Malaysia Ministry of Higher Education (MOHE)). The funders had no role in study design, data collection and analysis, decision to publish, or preparation of the manuscript.

==============================
Coffee beans derived from feces of the civet cat are used to brew coffee known as kopi luwak (the Indonesian words for coffee and palm civet, respectively), which is one of the most expensive coffees in the world owing to its limited supply and strong market demand. Recent metabolomics studies have revealed that kopi luwak metabolites differ from metabolites found in other coffee beans. To produce kopi luwak, coffee beans are first eaten by civet cats. It has been proposed that fermentation inside the civet cat digestive tract may contribute to the distinctively smooth flavor of kopi luwak, but the biological basis has not been determined. Therefore, we characterized the microbiome of civet cat feces using 16S rRNA gene sequences to determine the bacterial taxa that may influence fermentation processes related to kopi luwak. Moreover, we compared this fecal microbiome with that of 14 other animals, revealing that Gluconobacter is a genus that is, uniquely found in feces of the civet cat. We also found that Gluconobacter species have a large number of cell motility genes, which may encode flagellar proteins allowing colonization of the civet gut. In addition, genes encoding enzymes involved in the metabolism of hydrogen sulfide and sulfur-containing amino acids were over-represented in Gluconobacter. These genes may contribute to the fermentation of coffee beans in the digestive tract of civet cats.

Introduction

Coffee beverages have become the most popular staple drinks in many countries. Among different coffee varieties, kopi luwak is one of the most famous, in which the beans are collected from feces of the civet cat (Paradoxurus hermaphroditus). In many countries of Southeast Asia, including Philippines, Malaysia, and Indonesia, wild civet cats eat the beans produced by coffee trees (Patou et al., 2010). Once eaten, the outer layers of the beans are digested inside the gastrointestinal tract. When partially digested coffee beans are excreted in the feces, they are collected, cleaned, dried, and roasted. The roasted beans are then marketed as an expensive commodity (Jumhawan et al., 2016). It has been shown that the chemical and physical properties of kopi luwak beans are different from those of other coffees (Marcone, 2004). Indeed, a metabolomics approach showed that kopi luwak beans contain higher malic acid and citric acid contents and inositol-pyroglutamic acid ratios (Jumhawan et al., 2013). In another study, it has been shown that kopi luwak has lower levels of proteins but higher levels of lipids and carbohydrates than other coffees (Marcone, 2004).

Although the molecular mechanisms for these differences remain unknown, it is possible that the fermentation process inside the civet gut may influence the content of the kopi luwak metabolites. The unique gut microbiome of the civet cat might contribute to the characteristic physical properties and metabolite content of kopi luwak beans. A recent isolation and cultivation approach identified the bacterial species in the civet cat gastrointestinal tract, including Bacillus, Pseudomonas, Pantoea, Escherichia, Lactobacillus, Ochrobactrum and Kocuria (Suhandono et al., 2016). However, the study was unable to define the entire microbiome because some of the gut bacteria were unculturable (De Jager & Siezen, 2011). To address this shortcoming, we assessed the gut microbiome of the civet cat to identify the bacterial species that may contribute to the fermentation of kopi luwak beans using the 16S rRNA gene sequence. This approach allows us to detect unculturable bacteria. We collected three samples of civet cat feces from a Malaysian coffee farm. To characterize the civet cat fecal microbiome, we used the 16S rRNA gene sequence to determine the bacterial taxonomy. Our microbiome analysis of the 16S rRNA gene sequence revealed an abundance of Gluconobacter species. Our comparative genomics between Gluconobacter species, which are most similar to the Gluconobacter of the civet cat gut microbiome, revealed that the over-represented genes of certain sulfur metabolic pathways, such as sulfate reduction or cysteine and methionine metabolism, were also among the top-ranked modules/pathways. These pathways may contribute to the fermentation of kopi luwak beans in the gut.

Materials and Methods

Collection of fecal samples and DNA extraction

Three fresh fecal samples from wild civets were collected in the morning on June 20, 2015 from a coffee plantation (GPS coordinate: latitude: 1.7495934, longitude: 103.387647) in Johor, Malaysia. A field permit was not required since the raw kopi luwak specimens were provided by the owner of the coffee plantation, Mr. Jason Liew. The samples were kept at 4 °C for ≤2 h before being used for DNA extraction. To improve the DNA yield during extraction, ~0.2 g of each fecal sample was first incubated in 200 μl of suspension buffer (10 mM Tris-HCl, pH 8.0, 0.1 mM EDTA) containing 2 mg/ml lysozyme (Sigma Aldrich, St. Louis, MO, USA) at 37 °C for 30 min to remove the bacterial cell wall. The DNA was extracted from feces using the QIAmp® DNA stool kit (Qiagen, Hilden, Germany). To digest RNA, RNase A (Sigma Aldrich, St. Louis, MO, USA) was added (final concentration of 0.1 mg/ml) to buffer ASL from the stool kit. The quantity and quality of genomic DNA were estimated using a NanoDrop™ 2000/2000c spectrophotometer (Thermo Fisher Scientific, Waltham, MA, USA).

PCR amplification, sequencing and analysis of the 16S rRNA gene

A fragment of the 16S rRNA gene was amplified using the prokaryotic universal primer set 27F (5′-AGAGTTTGATCCTGGCTCAG-3′) and 338R (5′-TGCTGCCTCCCGTAGGAGT-3′) and Ex Taq DNA polymerase, hot-start version (Takara Bio, Shiga, Japan). PCR was carried out in 50-μl reaction volumes containing forward and reverse primers (1 μM each), 20 μl of template DNA, 5 μl of 10× Ex Taq Buffer, and 4 μl of dNTP mix from the Ex Taq kit. The PCR program consisted of 95 °C for 5 min followed by 31 cycles of 95 °C for 30 s, 53 °C for 30 s and 72 °C for 30 s, and a final extension at 72 °C for 3 min. The PCR products were individually concentrated and purified using a 2% E-Gel SizeSelect agarose gel (Thermo Fisher Scientific, Waltham, MA, USA), quantified using the Quant-iT dsDNA HS Assay kit (Thermo Fisher Scientific, Waltham, MA, USA), and qualified using the High Sensitivity DNA kit (Agilent Technologies, Santa Clara, CA, USA). Then, we checked a negative-control sample, in which PCR produced no visible bands on an agarose gel. Sequencing was carried out with the Ion PGM Sequencing 400 kit (Thermo Fisher Scientific, Waltham, MA, USA). Raw data for 16S rRNA gene sequences were analyzed with the VITCOMIC2 web application (http://vitcomic.org/) to derive the genus composition with the option “Conduct 16S rRNA gene Copy number normalization?—No”. VITCOMIC2 can estimate microbial community composition based on the sequence data for the 16S rRNA gene obtained from both metagenomic shotgun and amplicon sequencing (Mori et al., 2018). The sequence data are available in the DDBJ DRA database (https://www.ddbj.nig.ac.jp/dra/index-e.html) under accession DRA006640.

16S rRNA gene sequence–based phylogenetic analysis

The 16S rRNA gene sequences from bacteria in civet cat feces were checked using Trimmomatic version 0.33 with parameters “SE LEADING:17 TRAILING:17 AVGQUAL:25 MINLEN:200” to filter out low-quality sequences (Bolger, Lohse & Usadel, 2014). The three most abundant Gluconobacter sequences (termed civet cat004:Uniq1, civet cat005:Uniq1, civet cat006:Uniq1) were retrieved from each sample using the fastx unique command implemented in USEARCH version 10.0.240 (Edgar, 2010). To compare our Gluconobacter 16S rRNA gene sequences with those of other Gluconobacter species, we collected 16S rRNA gene sequences of Gluconobacter and Acetobacter from the SILVA Living Tree Project version 128 (Yilmaz et al., 2014). The sequences were aligned with MAFFT version 7.313 (Nakamura et al., 2018) and used as input to construct a phylogenetic tree using the neighbor-joining method with a Kimura 2-parameter model of nucleotide substitution using MEGA7 (Kumar, Stecher & Tamura, 2016). Acetobacter was used as an outgroup as the most similar taxonomic group to Gluconobacter species.

Collection of animal fecal 16S rRNA gene sequence data from other studies

Data for the bacterial genera identified in feces of various animals were collected from MicrobeDB.jp (MicrobeDB.jp Project Team, 2017). MicrobeDB.jp is a public database that stores the taxonomic composition data computed by VITCOMIC2 for most 16S rRNA gene sequences in the International Nucleotide Sequence Database Collaboration-Sequence Read Archive (INSDC-SRA), with integrated sampling site information such as environment and host categories. After downloading all the taxonomic composition data, all data with alignment counts less than 1,000 were filtered out. We listed the number of 16S rRNA gene sequence samples used in a comparative analysis after quality control of the raw sequence data (Table 1). Next, we converted alignment counts to mean relative abundance for each sample. After selecting samples that had an annotation of feces (MEO_0000054) by Metagenome and Microbes Environmental Ontology (MEO), we grouped samples into host categories and calculated their average relative abundances to serve as a reference microbiome composition for the gut of all animals in the analysis (MicrobeDB.jp Project Team, 2017).

Table 1 A list of animals and their total number of the 16S rRNA gene DNA sequence samples used in the comparative analysis.

Taxonomy	Animal	Number of fecal sample	Collected from	
Paradoxurus hermaphroditus	Civet cat	3	This study	
Anas platyrhynchos	Mallard duck	1	MicrobeDB.jp	
Anser sp	Anser bird	1	MicrobeDB.jp	
Bos taurus	Cattle	2	MicrobeDB.jp	
Canis lupus familiaris	Dog	69	MicrobeDB.jp	
Equus caballus	Horse	166	MicrobeDB.jp	
Gallus gallus	Chicken	1	MicrobeDB.jp	
Homo sapiens	Human	3003	MicrobeDB.jp	
Macaca fascicularis	Monkey	353	MicrobeDB.jp	
Mesocricetus auratus	Golden hamster	29	MicrobeDB.jp	
Mus musculus	Mouse	1346	MicrobeDB.jp	
Neotoma lepida	Desert woodrat	22	MicrobeDB.jp	
Rana pipiens	Frog	9	MicrobeDB.jp	
Sus scrofa	Wild boar	1	MicrobeDB.jp	
Sus scrofa domesticus	Pig	141	MicrobeDB.jp	

Comparative genomics of the major microorganisms in feces from civet cat and other animals

To provide a plausible explanation for the known metabolite profiles of kopi luwak, we compared the major bacterial species in civet cat feces (G. frateurii and G. japonicus) with the top nine dominant bacterial genera in feces from other animals (Bacteroides, Prevotella, Barnesiella, Lactobacillus, Oscillibacter, Citrobacter, Streptococcus, Faecalibacterium and Enterococcus). We downloaded complete amino acid sequences for each bacterial species from the National Center for Biotechnology Information (NCBI) RefSeq collection (Table S1) and annotated them with the reference prokaryote Kyoto Encyclopedia of Genes and Genomes (KEGG) orthology (KO) amino acid sequences (KEGG FTP Release 82.0) using DIAMOND version 0.9.3 with the parameter “blastp—max-target-seqs 5—e value 1E−2—id 70—min-score 40” (Buchfink, Xie & Huson, 2015). We used the top-hit results with e-values less than 1E−8 as the KO annotation for the amino acid sequences.

To compare KO terms, we calculated the median number of paralogs for each KO in each bacterial genus, including Gluconobacter from the NCBI RefSeq collection (Table S1). For the Gluconobacter genus, we used G. frateurii and G. japonicus because these two species are most closely related to the Gluconobacter species of the civet cat fecal microbiome (Table S2). To determine the KO(s) that was over-represented in Gluconobacter but not in major fecal microbial genera of other animals, we calculated the KO(i,ratio) as follows.

KO(i,ratio)=KO(i,Gluconobacter)KO(i,majorfecalmicrobialgenera)

Here, the KO(i,Gluconobacter) is calculated as the median for the number of KOi homologs in each Gluconobacter strain. Conversely, KO(i,majorfecalmicrobialgenera) is the mean of the median for the the number of KOi homologs in the top nine major bacterial genera in other animal feces (see Table S2). To estimate the over-represented functional units (KEGG module/KEGG pathway) in Gluconobacter (G. frateurii and G. japonicus), we performed Fisher’s exact test for each functional unit based on the number of over-represented KOs belonging to each functional unit in Gluconobacter. The KOratioi was converted to the log2 fold-change. We calculated (A) the number of KOs in a specific functional unit i, (B) the number of KOs out of the functional unit i, (C) the number of KOs with a log2 fold-change greater than 1 in specific functional unit i and (D) the number of KOs with a log2 fold-change greater than 1 in functional unit i. These four types of numbers for KO were used to perform Fisher’s exact test with R version 3.3.2 (R Core Team, 2019; Fisher, 1962).

Pvaluefunctionaluniti=Fisher′sexacttest[[A,B],[C,D]]

For a P value ≤ 0.05 with Fisher’s exact test, the generated functional unit i can be considered a special functional unit of G. frateurii and G. japonicus.

To identify functional units in G. frateurii and G. japonicus with log2 fold-changes > 1 relative to the fecal microbes from other animals, we mapped each functional unit (P value ≤ 0.05) and the positive ratio of G. frateurii and G. japonicus for each KO using the FuncTree2 web service (https://bioviz.tokyo/functree2/) (Uchiyama et al., 2015). This web service is a hierarchical visualization tool for KEGG pathways/modules/KO.

Results

Comparison of fecal microbial communities between civet cat and other animals

We analyzed the 16S rRNA gene sequences of microbes in each of three feces samples obtained from three different civet cats, which yielded 28,120,904 raw reads. Among all samples, the five most average relative abundant genera were Gluconobacter (66%), Citrobacter (14%), Acetobacter (7%), Enterobacter (2%) and Clostridium XI (1%) (Table S3). To determine the bacterial genus that is, predominant in the civet cat fecal microbiome, we compared the civet cat fecal 16S rRNA gene sequencing results with those obtained from the fecal microbiome of other animals. The data for the genus composition of the fecal microbiome of 14 animals were obtained from MicrobeDB.jp (MicrobeDB.jp Project Team, 2017) (http://microbedb.jp). Microbial composition data derived from the same computational pipeline were used for the civet cat fecal microbiome 16S rRNA gene sequence analysis. Among all fecal samples from all animals, the largest sample size was that of human feces (n = 3,003), with the second largest being mouse feces (n = 1,346). Most of the human feces samples have been acquired previously for large-scale projects such as the Human Microbiome Project (Turnbaugh et al., 2007) and the Metagenomics of the Human Intestinal Tract project (Ehrlich, 2011) (Table 1).

We determined the microbial genus composition of feces for each animal species by averaging the relative abundance among all fecal samples for each animal. A comparison of these derived compositions with that determined for civet cat feces revealed a marked difference in bacterial genus composition (Fig. 1; Table S3). For example, Gluconobacter was found to be the predominant genus, representing 60% of the bacterial abundance in all civet cat samples. By contrast, Gluconobacter was rarely detected in the fecal microbiomes of the 14 animals used in the other studies. Therefore, civet cat apparently has a unique fecal microbiome compared with other animals.

Figure 1 Hierarchical clustering of bacterial genera of microbes identified in feces from different animals.

The relative abundance of each bacterial genus was estimated for the fecal microbiome of various animals. The microbial composition in feces from each animal is presented as the relative abundance. The 10 most abundant genera observed in animal feces are labeled with different colors; other genera are labeled “others”. The hierarchical clustering tree was generated based on the Euclidean distance between the bacterial genus level compositions.

Phylogenetic analysis of Gluconobacter species in civet cat feces

To determine the Gluconobacter species that are most common in the civet cat fecal microbiome, we created a phylogenetic tree for Gluconobacter species and Acetobacter species as the outgroup in civet cat fecal samples (Fig. 2). The tree showed that Gluconobacter comprises two groups, namely group 1 and group 2. The species in civet cat fecal samples belonged to group 2 (Fig. 2). The tree also showed that the Gluconobacter species in the civet cat fecal microbiome belonged to the group including G. cerinus, G. frateurii, G. japonicus, G. nephelii, G. thailandicus and G. wancherniae (Fig. 2; Fig. S1). To determine the KO terms for Gluconobacter species, we calculated the KO terms for the amino acid sequences of 39 Gluconobacter strains from the Reference Sequence (RefSeq) database of NCBI. We found that RefSeq genomes for G. nephelii and G. wancherniae were not available in NCBI; hence, these two species were not included in the KO analysis. Next, using hierarchical clustering of the KO terms for each Gluconobacter strain, we showed that strains with similar 16S rRNA gene sequences clustered together, with distinct separations among the clades (Fig. 3). These results demonstrated that the Gluconobacter species of the civet cat fecal microbiome were most likely to be G. frateurii and G. japonicus, and that the KO terms of these two species differed from those of other Gluconobacter species, especially species in Gluconobacter group 1.

Figure 2 Phylogenetic tree for the Gluconobacter genus.

The phylogenetic tree was based on 16S rRNA gene sequences (27F-338R) and was constructed using the neighbor-joining method. The Acetobacter species were used as an outgroup (green). Data for Gluconobacter group 1 (orange) and group 2 (purple) were from the SILVA Living Tree Project version 128. The Gluconobacter sequences identified in the civet cat fecal samples are clustered together (black) with group 2.

Figure 3 KO terms of each Gluconobacter strain from the RefSeq database.

Gluconobacter strains within each lineage had similar KO terms. Rows represent the 39 Gluconobacter strains, and columns represent the 1,653 KO terms that were identified in more than one strain. The strains were clustered based on Euclidean distances calculated from the KO terms for each Gluconobacter strain. The presence of a KO term is indicated by yellow highlighting, and absences are indicated by black. Each Gluconobacter strain is colored based on the phylogenetic analysis presented in Fig. 2, that is, Gluconobacter group 1 is shown in orange, Gluconobacter group 2 is shown in purple, and unknown species of Gluconobacter are not colored.

Comparative genomics of genera of the feces microbiome

We next compared the KO terms for G. frateurii and G. japonicus, which are the most similar species to the Gluconobacter species of the civet cat fecal microbiome, with the KO terms of the dominant genera in feces of other animals. Other dominant bacterial genera in the feces of other animals were Bacteroides, Prevotella, Barnesiella, Lactobacillus, Oscillibacter, Citrobacter, Streptococcus, Faecalibacterium and Enterococcus (Fig. 1). This comparative genomics analysis provided a list of candidate microbial genes that may be involved in the fermentation of kopi luwak in the civet cat digestive tract.

To identify the KO terms of each genus, we downloaded the amino acid sequences of RefSeq genomes for these genera (Table S1). These sequences were mapped to prokaryote KO amino acid sequences that are publicly available in the KEGG. The KEGG database contains information about most of the bacterial genome sequences (Kanehisa, 2000). Finally, we calculated the median number of KOs for each genus. Likewise, we calculated the median number of KOs for G. frateurii and G. japonicus because these two species were the predominant Gluconobacter species of the civet cat fecal microbiome (Fig. 4; Table S2). Interestingly, certain KO terms were more abundant in G. frateurii and G. japonicus than in other Gluconobacter species or were present in only these two species. We next determined the KO terms that were enriched in G. frateurii and G. japonicus compared with other major genera of the various animal fecal microbiomes in our analysis. We calculated log2 fold-change values, which were derived from the median number of KO terms of G. frateurii and G. japonicus per mean number of KO terms of other dominant bacterial genera in the feces of other animals; this allowed us to determine enrichment, and we counted the number of KO terms with log2 fold-change greater than 1.0. These results showed that at least 664 KOs of the 3,623 total had a log2 fold-change value ≥ 1.0 (Table S4). To minimize false-positive results, we performed Fisher’s exact test for each KEGG module and pathway (see “Methods”). The following pathways and modules were found to be more abundant in G. frateurii and G. japonicus than in other animal fecal microbes: porphyrin and chlorophyl metabolism (map00860); cobalamin biosynthesis, cobinamide => cobalamin (M00122); flagellar assembly (map02040); methionine salvage pathway (M00034); histidine biosynthesis, PRPP => histidine (M00026); assimilatory sulfate reduction, sulfate => H2S (M00176); cysteine and methionine metabolism (map00270) (Fig. 5). Moreover, we analyzed over-represented KO terms of the TCA cycle (map00020) in G. frateurii and G. japonicus because a previous study demonstrated that the levels of malate and citrate produced from the TCA cycle are higher in kopi luwak beans than other coffee beans (Jumhawan et al., 2016). Although Fisher’s exact test could not detect any differences in the KOs for the TCA cycle among different coffee beans (map00020), this test showed that certain KOs of the TCA cycle (map00020) were over-represented in G. frateurii and G. japonicus (Fig. 5; Table S2).

Figure 4 Hierarchical clustering of bacterial genera based on the number of paralogs for each KO term.

The genus and Gluconobacter species were clustered based on Euclidean distances calculated from each median number of paralogs for each of the KO terms (3,627). The color scale of the heat-map indicates the number of paralogs for each KO term.

Figure 5 The characteristic KO terms for G. frateurii and G. japonicus in FuncTree2.

The orange circle is the negative-log–scaled P value obtained from Fisher’s exact test. The green bar is the positive ratio of G. frateurii and G. japonicus. The KO terms of the TCA cycle are mapped at the top left of the figure.

Discussion

In this study, we compared the civet cat fecal microbiome with microbiomes that have been established for 14 other animals, including humans. In the civet cat fecal microbiome, Gluconobacter was found to be the most abundant genus. To elucidate the relationship between genetic factors and microbiome composition, we checked public resources concerning the genetic contents of the gut microbiome genera in the family Felidae, which is known to be the closest taxon to the civet cat (Nyakatura & Bininda-Emonds, 2012). Two studies described the microbiome of the domesticated cat, genus Felidae. However, these studies did not report the presence of Gluconobacter in the samples (Deusch et al., 2014, 2015).

Another study reported that the human fecal microbiome differs among countries, suggesting that various diets might be able to explain the difference (Moeller et al., 2014; Kartzinel et al., 2019). It has been suggested that different diets may help to explain the uniqueness of the civet cat fecal microbiome, which may be influenced by uncommon foods such as coffee beans. The most important point is that previous studies of the animal fecal microbiome did not detect a microbiome dominated by Gluconobacter.

Moreover, our phylogenetic analysis demonstrated that Gluconobacter species that are closely to G. frateurii, G. japonicus, G. nephelii, and G. wancherniae are the most abundant species in the civet cat fecal microbiome. A previous study reported that G. frateurii, G. japonicus are closest to each other (Malimas et al., 2009). Our study suggests that the civet cat fecal microbiome is dominated by Gluconobacter, which was detected in a previous study using general cultivation methods (Suhandono et al., 2016). In fact, a previous study of the cacao microbiome reported that Gluconobacter species is hardly detected by the cultivation method; however, using metagenomics and shotgun sequencing, the presence of Gluconobacter species was observed (Agyirifo et al., 2019).

Some bacteria can grow within only a narrow temperature range. To determine the optimum growth temperature of various Gluconobacter species, we retrieved information about the growth temperature of Gluconobacter (NITE, 2018). Most Gluconobacter species grow at 37 °C, which is the body temperature of the civet cat (Fig. S2). Interestingly, upon their initial sampling from feces, fresh coffee cherries contain a Gluconobacter sp. (De Bruyn et al., 2017). Although we could not analyze the microbiome from fresh coffee cherries prior to digestion, these data support the notion that Gluconobacter of civet cat feces may originate from fresh coffee cherries and that species of this genus can grow at 37 °C in the gastrointestinal tract of civet cat and may be involved in the fermentation process of kopi luwak. Interestingly, Gluconobacter species are obligate aerobic microorganisms (Barberán et al., 2017); however, a recent study mentioned that oxygen is rich in the small intestine of humans. Likewise, the civet cat intestine may contain some oxygen that could facilitate Gluconobacter fermentation in the civet cat intestine (Berean et al., 2018).

We identified some over-represented KEGG modules or pathways that may influence the physical and chemical properties of kopi luwak. In this study, we focused on KOs that appeared more frequently in G. frateurii and G. japonicus than in the major genera of fecal microbiomes of other animals (Fig. 5). Interestingly, flagellar assembly (map02040) is one of the characteristic pathways in Gluconobacter. A recent study of fecal microbiomes revealed that bacterial flagellar proteins promote the production of flagellum-specific immunoglobulin derived from the host, and this immunoglobulin type inhibits bacterial motility (Cullender et al., 2013). Although the bacterial flagellum may cause inflammation in the mammalian gut, civet cats may have a tolerant immunity to the flagella of Gluconobacter. In addition, the bacterial flagellum often helps colonize the attachment and invasion of bacteria to the host cell (Haiko & Westerlund-Wikström, 2013). Hence, flagellum of Gluconobacter may be related to colonization of them in the civet cat gut. Conversely, the methionine salvage pathway (M00034); histidine biosynthesis, PRPP => histidine (M00026); assimilatory sulfate reduction, sulfate => H2S (M00176); and cysteine and methionine metabolism (map00270) were also among the top-ranked modules/pathways. Sulfur-containing compounds are generally volatile and thus can affect the flavor and aroma of many foods (Reineccius, 2005). A previous study has shown that G. frateurii can produce H2S from thiosulfate (Lori & Claus, 1989). Although it has been shown that sulfur-containing compounds contribute to the aroma of coffee (Czerny, Mayer & Grosch, 1999), it remains to be determined whether the content of sulfur-containing compounds in kopi luwak differ from those of other coffees. In a previous study, Gluconobacter was detected in the fermentation of cacao samples using metagenomics analysis, and this bacteria is implicated in the fermentation process (Agyirifo et al., 2019). Thus, Gluconobacter might contribute to the production of kopi luwak.

Traditional metabolomics approaches have identified citric acid and malic acid as discriminative metabolites for kopi luwak and other coffees (Jumhawan et al., 2013, 2016). Although the TCA cycle (map00020) was not over-represented and statistically significant in our study, it is one of the pathways involved in the production or consumption of malic acid and citric acid. Consistent with this phenomenon, we observed some KO terms with a relatively large number of homologs in Gluconobacter compared with other bacteria in the TCA cycle (Fig. 5). Therefore, the TCA cycle activity in Gluconobacter species in the civet cat gut may promote the observed increased production of malic acid and citric acid in kopi luwak. Interestingly, the TCA cycle of Gluconobacter spp. is incomplete due to the lack of succinate dehydrogenase for the production of fumarate and succinate (Prust et al., 2005). As a result, the incomplete TCA cycle may also help to boost the production of citric acid and malic acid.

Overall, our results reveal biological factors that may be involved in the fermentation of kopi luwak in the gut of civet cat. Our current study focuses on only G. frateurii and G. japonicus, but other civet cat fecal microbes, such as Citrobacter and Clostridium, may also help ferment kopi luwak. To gain a more complete understanding of the biological factors involved in the fermentation of kopi luwak in the civet cat gut, a metagenomic analysis should be performed to investigate the functional genes that are highly expressed in most or all bacteria present in the civet cat fecal microbiome.

Conclusion

In this study, we aimed to characterize the fecal microbiome of civet cats. We found, for the first time, that the fecal microbiome of civet cats was dominated by Gluconobacter, namely G. frateurii, G. japonicus, G. nephelii and G. wancherniae, using the 16S rRNA gene sequence. Moreover, we also identified Gluconobacter species as having the large number of genes involved in cell motility and hydrogen sulfide and sulfur-containing amino acids, which may influence the fermentation of kopi luwak in the gut of civet cat.

Supplemental Information

Supplemental Information 1 Alignment of 16S rRNA genes from Gluconobacter species.

Click here for additional data file.

Supplemental Information 2 Maximum growth temperature of Gluconobacter strains.

Click here for additional data file.

Supplemental Information 3 Tables S1–S4.

Table S1. The RefSeq genomes list of each bacterial genus used for comparative genomics

Table S2. The number of KO paralogs generated with median of bacterial genus strains

Table S3. Microbial composition of each animal fecal samples

Table S4. The statistical value of KO paralogs number.

Click here for additional data file.

We would like to thank Mr. Jason Liew of My Liberica Coffee Plantation for providing raw kopi luwak specimens, and Dr. Yumiko Kurokawa for technical assistance with sequencing.

Additional Information and Declarations

Competing Interests

Author Contributions

Data Availability

The authors declare that they have no competing interests.

Hikaru Watanabe performed the experiments, analyzed the data, prepared figures and/or tables, authored or reviewed drafts of the paper, and approved the final draft.

Chong Han Ng performed the experiments, authored or reviewed drafts of the paper, and approved the final draft.

Vachiranee Limviphuvadh conceived and designed the experiments, authored or reviewed drafts of the paper, and approved the final draft.

Shinya Suzuki analyzed the data, prepared figures and/or tables, and approved the final draft.

Takuji Yamada conceived and designed the experiments, authored or reviewed drafts of the paper, and approved the final draft.

The following information was supplied regarding data availability:

The sequences are available at NCBI: PRJDB6640, the RefSeq genomes are listed in Table S1.

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
