# Peer review of "Gluconobacter dominates the gut microbiome of the Asian palm civet Paradoxurus hermaphroditus that produces kopi luwak"

_PeerJ, doi:10.7717/peerj.9579_

## Round 0.1 · original submission · Major Revisions

Dear Dr. Watanabe and colleagues:

Thanks for submitting your manuscript to PeerJ. I have now received three independent reviews of your work, and as you will see, the reviewers raised some concerns about the research. Despite this, these reviewers are optimistic about your work and the potential impact it will have on research studying the microbiome of the kopi luwak-producing civet. Thus, I encourage you to revise your manuscript, accordingly, taking into account all of the concerns raised by both reviewers.

While the concerns of the reviewers are relatively minor, this is a major revision to ensure that the original reviewers have a chance to evaluate your responses to their concerns. There are many suggestions, which I am sure will greatly improve your manuscript once addressed.

Importantly, please ensure that an English expert has edited your revised manuscript for content and clarity. Please also ensure that your figures and tables contain all of the information that is necessary to support your findings and observations, including complete legeneds. Please also make sure that all taxon names are spelled correctly, properly italicized, etc.

Please avoid make overstatements regarding your conclusions, ensuring that your data support your observations and interpretations.

I look forward to seeing your revision, and thanks again for submitting your work to PeerJ.

Good luck with your revision,

-joe

Reviewer 1 ·

Basic reporting

The English language should be improved in some sections to ensure the clarity of your text. Some examples are given in the different sections.

Figures:
fig 1 - the hierarchical clustering tree was generated based on bacterial genera? please clarify.
fig 2 - bacterial genus should be in italic. please check all the figures.
fig 5 - the membrane transport differences are not specified as the remaining ones. please do so.

Additional comments:
Line 49 - replace "fat" with " lipids"; carbohydrates (plural?)
line 50 - the mechanism responsible for the relatively higher levels of lipids and carbohydrates, and the lower levels of protein are known? if so please elaborate on this topic.
line 64 - specify the sulfur metabolic pathway

Experimental design

Some concerns can arise from the low number of samples (n=3). More data analysis should be performed to overcome this gap, such as perform prediction analysis regarding the functionality of gut microbiota (e.g. using PiCRUST https://doi.org/10.1038/nbt.2676).

Additional comments:
line 91 - high sensitivity DNA kit was used to quantified or assess the amplicons quality? please clarify.
line 95 - more information should be provided regarding VITCOMIC2
line 108 - why these agglomeration methods (neighbor-joining and kimura-2-parameter)? please clarify
line 127 - change "microbes" to "microorganisms"
line 149 - please rephrase
line 161 - specify what is the "a", "b", "c", and "d"

Validity of the findings

Some conclusions are excessive: line 211 - The KO terms of the civet cat Gluconobacter strains is not produced and consequently can not be analyzed, so, the only conclusion related to species identification can only be made based on 16S rRNA data, that only allows to state that the most probable species identification of the detected sequences is G. frateurii, G. japonicus, G. nephelii, or G. wancherniae. Thus, the exclusion of G. nephelii, or G. wancherniae as possible identification is not understandable.

More discussion: line 195 - it will be interesting to discuss the hierarchical tree produced using bacterial genera information to each host since birds and mammals gut microbiome seems to not differentiate (e.g. of studies reporting differences and non-differences https://doi.org/10.1038/nrmicro1978 + https://doi.org/10.1128/mBio.02901-19); a PCoA can be performed to better despite these differences and similarities.
line 289 - no real discussion is made regarding these pathways

Additional comments:
line 175 - indicate the relative abundance of each genus
line 146 - the comparison with the fecal microbiome of other animals was to determine the differentially detected genera in the civet cat and NOT to determine the predominant bacterial genus. please rectify.
line 181 - remove "were" and "our"
line 182 - remove the s in "sequences"
line 182 - add "analysis" at the end of the sentence.
line 191 - change "all species" to "bacterial abundance"
line 268 - add "these" between "that" and "two"
line 281 - change "indigestion" to "digestion" or "ingestion"
line 310 - separate "and" from "other"
line 313 - please state that the enrichment is not statistically significant
line 333 - change "which is the same group with" to "namely"

Additional comments

The English language should be improved in some sections to ensure the clarity of your text. More data analysis should be performed to overcome this gap, such as perform prediction analysis regarding the functionality of gut microbiota. Some conclusions are excessive and more discussion is needed.

·

Basic reporting

The English used is clear, the tables and figures are satisfactory. All other requirements in this section were met by the authors

Experimental design

Due to the comparing nature of civet gut microbiome to other animal gut microbiome, it will be appropriate to let it reflect in the title. Refer to line 252/253

Further, explain lines 57-59 for clarity on the statement of the problem.

The general methods are well written, however, there should be strict adherence to sections. such as methods should contain only methods, Results in only results and discussions only discussions (lines 60, 62-65, 277 etc).

Validity of the findings

The results reflect a critical evaluation of civet faeces microbiome.

The data DRA006640 is well deposited in DDBJ DRA database. It is mentioned the supplementary information by authors but not mentioned the manuscript. I expect to see such statements in methods
They are statistical sound.

The conclusions are sound and reflect the objective.

The article is generally good but needs the reorganisation into the appropriate sections.

Additional comments

This research is very interesting to read.

Reviewer 3 ·

Basic reporting

In the manuscript Yamada and coworkers nicely present their data obtained by 16S rRNA gene sequencing of palm civet cat feces and a bioinformatics downstream analysis. It was intended to determine the bacterial taxa that may influence fermentation processes of coffee beans in the gut of palm civet cats. These coffee beans collected from feces are used to brew coffee known as kopi luwak, one of the most expensive coffees in the world. The study revealed that Gluconobacter species appear to largely dominate in the gut microbiome of the Asian palm civet Paradoxurus hermaphroditus.
The manuscript is well written in clear, unambiguous, professional English language. The introduction provides the context and the relevant literature appear to be well referenced. The figures are mostly helpful and of sufficient quality. For Figures S1 and S2 no legend explaining the figure was present.

The availability of the Ion PGM sequencing raw data was not mentioned and the data appear to be not available.

Experimental design

The study presented reflects original primary research within aims and scope of the journal. The research questions are well defined, relevant and meaningful. It is outlined how the research fills an identified knowledge gap. The methods are described with sufficient details and information to replicate.

Validity of the findings

Finding Gluconobacter as dominating DNA is very surprising in view of the gut microbiomes of other animals and in view of the conditions typically anticipated in the gut. This is very likely related to the diet and the natural habitats of Gluconobacter species as mentioned by the authors, since Gluconobacter typically is present in blossoms and their fruits. However, the underlying DNA sequencing data appear to be not provided / publically available. Gluconobacter is strictly aerobic and would actually not be expected to actively contribute to fermentation processes in the gut. Therefore, the authors speculate that other microorganisms possibly are also involved in the fermentation of kopi luwak. It requires further studies to clarify this and the related metabolism potentially involved in the gut fermentation processes of kopi luwak beans. Therefore, the described results are very interesting and publication of the study is expected to stimulate further basic and applied research in this field in several ways.

Additional comments

This reviewer has only a few minor comments which are intended to improve the manuscript.

For the reader it would be interesting to read about the conditions at the coffee plantation and the potential diets of the Asian palm civet used in this study. Are there also any other fruits the animals could have ingested in relevant time when crossing or temporarily leaving the coffee plantation area? If not leaving the plantation area, what else than coffee fruits contribute to the diet? How long (days / hours) does it typically take after ingestion until the beans leave the gut?

As generally recognized in the literature and also mentioned by the authors, acetic acid bacteria including Gluconobacter are obligate aerobic microorganisms. Therefore, it appears to be difficult to accept Gluconobacter as an biologically active gut microorganism. However, in recent studies published by Kalantar-Zadeh ingestible sensors were developed and applied in human gut. Although the oxygen sensor capsule is not specific for oxygen (but oxidative agents), in a human study oxygen sensor-related data suggested an unexpected high content of oxygen in the small intestine after the stomach (PMID 30067289, Figure 1). This study could be included in the discussion by the authors, as this could probably also be the case in animals including Asian palm civet. The presence of higher oxygen concentrations in the intestine would increase the acceptance that Gluconobacter could be biologically active and contribute to fermentation of kopi luwak in a part of the civet cat gut. Furthermore, if not already applied or planned by the authors in further studies, mentioning Kalantar-Zadeh’s small ingestible sensors could promote its usage by other groups to study animal gut conditions in gut microbiome research (this reviewer is not related to Kalantar-Zadeh or his group etc.). This could be included in the discussion by mentioning the content of line 324 already in the paragraph in lines 281-284. Otherwise the important information that Gluconobacter species are obligate aerobic microorganisms will appear only at the very last to the readers that are not familiar with Gluconobacter.

L33 and L335
Which unique cell-motility gene has been identified by the authors in the Gluconobacter species? The name of the motility gene and the function of its protein should be mentioned in the manuscript. Furthermore, it should be explained why the authors consider this cell-motility gene to be unique.

L314
The wording “upregulation of the TCA cycle“ appears to be misleading in this context and should be modified.

Figures S1 and S2
Where are the legends of Figures S1 and S2 (not found)?

The Ion PGM sequencing raw data should be made publically available and this availability shoould be linked in the manuscript.

---

## Round 0.2 · accepted · Accept

Dear Dr. Watanabe and colleagues:

Thanks for revising your manuscript based on the concerns raised by the reviewer. I now believe that your manuscript is suitable for publication. Congratulations! I look forward to seeing this work in print, and I anticipate it being an important resource for groups studying the microbiome of the kopi luwak-producing civet. Thanks again for choosing PeerJ to publish such important work.

Best,

-joe

Reviewer 1 ·

Basic reporting

English has been improved. Background information and figures have been improved.
The reviewer suggests the inclusion of the NMS analysis, present in the rebuttal letter, in the final manuscript (at least as a supplementary figure).

Experimental design

The authors addressed the experimental design concerns of the reviewer.

Validity of the findings

The authors addressed the speculation concerns of the reviewer.

Additional comments

The authors correctly addressed all reviewer concerns and modified the final manuscript accordingly. The addition of the NMS analysis in the final manuscript would support some of the authors' results/conclusions.

·

Basic reporting

The English used is clear, the tables and figures are satisfactory.
All other requirements in this section were met by the authors

Experimental design

I agree to the explanation given to the suggestion on the article title.

Validity of the findings

Based on the standard methods used in the research, the results obtained are valid. The authors have now stated the availability of data and the database in the manuscript.

Reviewer 3 ·

Basic reporting

no comment

Experimental design

no comment

Validity of the findings

no comment

Additional comments

no comment